# Map Reading and Analysis with GPT-4V(ision)

**Jinwen Xu** [1] **and Ran Tao** [2,*]

1    GIS Center, Florida International University, Miami, FL 33199, USA; jinwxu@fiu.edu
2    School of Geosciences, University of South Florida, Tampa, FL 33647, USA
*    Correspondence: rtao@usf.edu

**Abstract:** In late 2023, the image-reading capability added to a Generative Pre-trained Transformer (GPT) framework provided the opportunity to potentially revolutionize the way we view and understand geographic maps, the core component of cartography, geography, and spatial data science. In this study, we explore reading and analyzing maps with the latest version of GPT-4-vision-preview (GPT-4V), to fully evaluate its advantages and disadvantages in comparison with human eye-based visual inspections. We found that GPT-4V is able to properly retrieve information from various types of maps in different scales and spatiotemporal resolutions. GPT-4V can also perform basic map analysis, such as identifying visual changes before and after a natural disaster. It has the potential to replace human efforts by examining batches of maps, accurately extracting information from maps, and linking observed patterns with its pre-trained large dataset. However, it is encumbered by limitations such as diminished accuracy in visual content extraction and a lack of validation. This paper sets an example of effectively using GPT-4V for map reading and analytical tasks, which is a promising application for large multimodal models, large language models, and artificial intelligence.

**Keywords:** GPT; LMMs; map reading; map analysis; spatial patterns

## 1. Introduction

Maps are the core of cartography, a discipline that encompasses the conception, production, dissemination, and study of these essential representations of space [1]. They are fundamental tools for understanding our world, encapsulating complex spatial information in an accessible visual format. The evolution of cartography reflects the ever-changing human understanding and interaction with the physical environment [2,3]. The importance of map reading and analysis cannot be overstated; it extends beyond navigation and geographical orientation. Maps are critical in a wide range of fields, from urban planning [4,5] and environmental management [6] to geopolitics [7] and disaster response [8]. The skill of map reading and analysis is important in interpreting these representations accurately, enabling users to extract meaningful insights from complex spatial data. Maps are not just literal guides but layered with context; they tell stories of land use, demographic trends, and socio-economic patterns. In essence, maps are a blend of science, art, and storytelling. Through careful analysis, maps can reveal unspoken histories and offer unique insights into historical and current societal trends, helping us comprehend spatial relationships and patterns [9]. With the advent of digital cartography and Geographic Information Systems (GIS), maps have become more dynamic and interactive, allowing for a deeper, more analytical engagement with spatial data [10].

The era of big data has simultaneously enriched and complicated map reading and map analysis. While visual inspection of maps remains an indispensable research step for recognizing spatial patterns and understanding physical environments, the sheer volume of data now available poses significant challenges. Traditional map reading and analysis techniques often fall short in handling the massive and complex datasets that need to be mapped and analyzed [11]. To address these challenges, various advanced technologies have been deployed. These include web mapping and mapping with Artificial Intelligence

(AI), but there is a notable gap in technologies aimed at automating the process of map reading and analysis [12,13]. Previously, techniques like Optical Character Recognition (OCR) were employed in automating these tasks, but they often struggled to accurately extract complex features from maps, limiting their application in this field [14]. AI, specifically in the domain of image recognition, is another potential alternative, but its effectiveness in map reading and analysis is contingent on the availability of large, high-quality training datasets.

The introduction of advanced Large Language Models (LLMs) and Large Multimodal Models (LMMs) marks a significant milestone in the evolution of artificial intelligence and machine learning [15]. The Generative Pre-Trained Transformer (GPT) series, developed by OpenAI, stands as a flagship example of LLM development in the field of natural language processing and AI [16]. GPT models have been applied in a variety of domains and can perform various types of tasks, such as creative writing [17], language translation [18], extraction of location descriptions from social media [19], making maps [12], coding assistance [20], and data science [21]. When additional modalities (such as image inputs) are integrated, LLMs become a powerful visual language model and bring more functionalities. With the release of the frontier LMM, GPT-4-vision-preview (GPT-4V(ision), abbreviated as GPT-4V), on 6 November 2023, a new horizon has opened up [22]. In a recent evaluation report of GPT-4V, researchers found that GPT-4V has superior abilities to describe images, localize and count objects, provide dense captioning, exhibit multimodal knowledge and commonsense reasoning, analyze scene text, tables, charts, and documents, understand multilingual and multimodal content, and integrate coding capabilities with visual understanding [23]. These outstanding capabilities bring potential advantages such as automating map reading across various spatial and temporal scales, and identifying distribution patterns that may elude human observation, such as complicated point patterns. Additionally, GPT-4V relates findings from maps to its extensive knowledge base, enhancing interpretive capabilities. For researchers, there emerges an opportunity to use GPT-4V as a digital assistant for map analysis. For students, GPT-4V can serve as an educational tool, teaching map reading skills. Moreover, it can help non-experts interpret domain-specific maps, making complex information accessible to a broader audience. Such characteristics of GPT-4V present a promising opportunity to further explorations and refine the use of AI in map reading and analysis.

This study serves as a pilot work to systematically explore and evaluate GPT-4V's capabilities in map reading and analysis. Our investigation includes two sections: first, we test GPT-4V's proficiency in retrieving information from the map content, legend, scale, colors, symbols, labels, and other map elements, in comparison with two other LMMs (baselines); second, we assess its effectiveness in common map analysis, including recognizing spatial distribution patterns, e.g., point pattern and bivariate point pattern recognition, and analyzing the differences between maps with (1) same spatial scale but different temporal scales and (2) the same temporal scale but different spatial scales. Considering that there are some other competitive LMMs developed by other companies, such as Gemini Pro Vision by Google Inc. (USA) and Sphinx by OpenGVLab, we conducted a comparison between the three LMMs regarding their map reading abilities. However, due to a lack of functionality to handle multiple images, Gemini Pro Vision and Sphinx can only read one image at a time, making it difficult to compare several images. Thus, in the second part (map analysis), we focus on testing GPT-4V's capability. In summary, our approach aims to test GPT-4V in reading and analyzing a series of sample maps from various sources through supported API. Through these experiments, we seek to provide a comprehensive assessment of GPT-4V's capabilities in this uncharted territory. We conclude with our findings about how it may potentially revolutionize the way we examine geographic maps, as well as handling and understanding spatial data.

## 2. Map Reading

Professional digital maps are created through a process of data selection, classification, generalization, and symbolization to illustrate geographic information in a certain area [24]. Users interpret these maps through a process akin to information retrieval, extracting and understanding data from the cartographer's visual representation. The map reading process typically involves the detection of symbols, their discrimination, understanding their meanings, recognizing these symbols, interpreting them by adding meaning, and retaining their relationships [25,26]. In the past, challenges in OCR and image processing hindered the extensive use of these techniques for map reading due to the complexity of map features. However, advancements in LLMs, LMMs, and AI have now made it feasible. For instance, GPT-4V, representing the next evolution in this field, integrates multi-sensory skills to surpass the general intelligence of traditional LLMs, such as GPT-4 (no vision) [23]. Referring to multiple references and previous literature [24,26–29], we conducted several map reading experiments on GPT-4V to evaluate its capability in recognizing various map elements, including legends, symbols, and spatial scales, and in comparing different types of domain-specific maps.

Specifically, Figure 1 illustrates the process of using GPT-4V to read map images and make corresponding responses. Similarly, Gemini Pro API (Gemini Pro Vision, version 1.0, through generativeai package) and Sphinx API (http://llama-adapter.opengvlab.com, accessed on 20 February 2024 through gradio_client package) were used to test the map reading abilities of these LMMs. We opted for the GPT-4 API over the ChatGPT interface on the OpenAI webpage due to its ability to simultaneously recognize multiple images and monitor processing times. This study focused on evaluating the textual responses from GPT-4V (currently limited to text-only outputs). Despite Dall-E 3, OpenAI's generative AI model on image generation, having the capability to produce image outputs, its limited image recognition abilities restrict its use in map reading and analysis [30], especially when compared to GPT-4V. Our experiments included a variety of map images sourced from both manually created maps, which were subsequently hosted and shared publicly on a cloud drive, and online map images from different hosting servers. Leveraging the fact that the GPT-4 API does not store uploaded files but can process image URLs, we chose Google Drive to host the tested images and created URLs accordingly. In the message sent to GPT API, the role parameter determines which contents should be pre-set in the model and which contents are user requests. The input information for GPT-4V in a request includes textual instructions (prompt) and images. In this study, we employed several types of requests to assess GPT-4V's capabilities in map reading, which involves information retrieval, and in map analysis, which pertains to visual analytics.

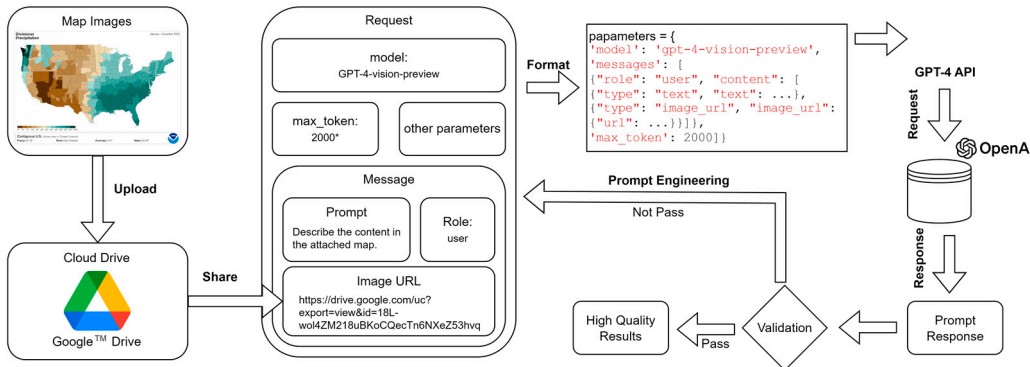

**Figure 1.** Map image reading and analysis process using GPT-4V in OpenAI API.

We also carefully considered the prompt engineering process before inputting our textual prompts according to the guidelines given by OpenAI that prompts should be clear, detailed, and structured [31]. We made an example of Prompt 2.1 and 2.2 to illustrate how different prompts can lead to a different answer. However, since most of the tasks tested in

the first section are straightforward and do not involve complexity, prompt engineering did not significantly improve the response from LMMs (see Figure S1).

### 2.1. Map Element Recognition

We first tested the ability of LMMs in map element recognition using the prompt below. Figure 2 is a map from the book '*Map use: reading, analysis, interpretation*' [24], which predicts the habitat of owls in Oregon. It includes a range of common map elements, such as legend, scale bar, title, subtitle, sources/credits, and figure caption. Specifically, the map illustrates the spotted owl in Oregon using a sequential color scheme to represent predicted habitat from None to Good. Two line types were used to distinguish ecoregional boundaries from county boundaries. By phrasing our prompt as shown below to test the ability of LMMs to recognize map elements in Figure 2, we obtained the response as follows.

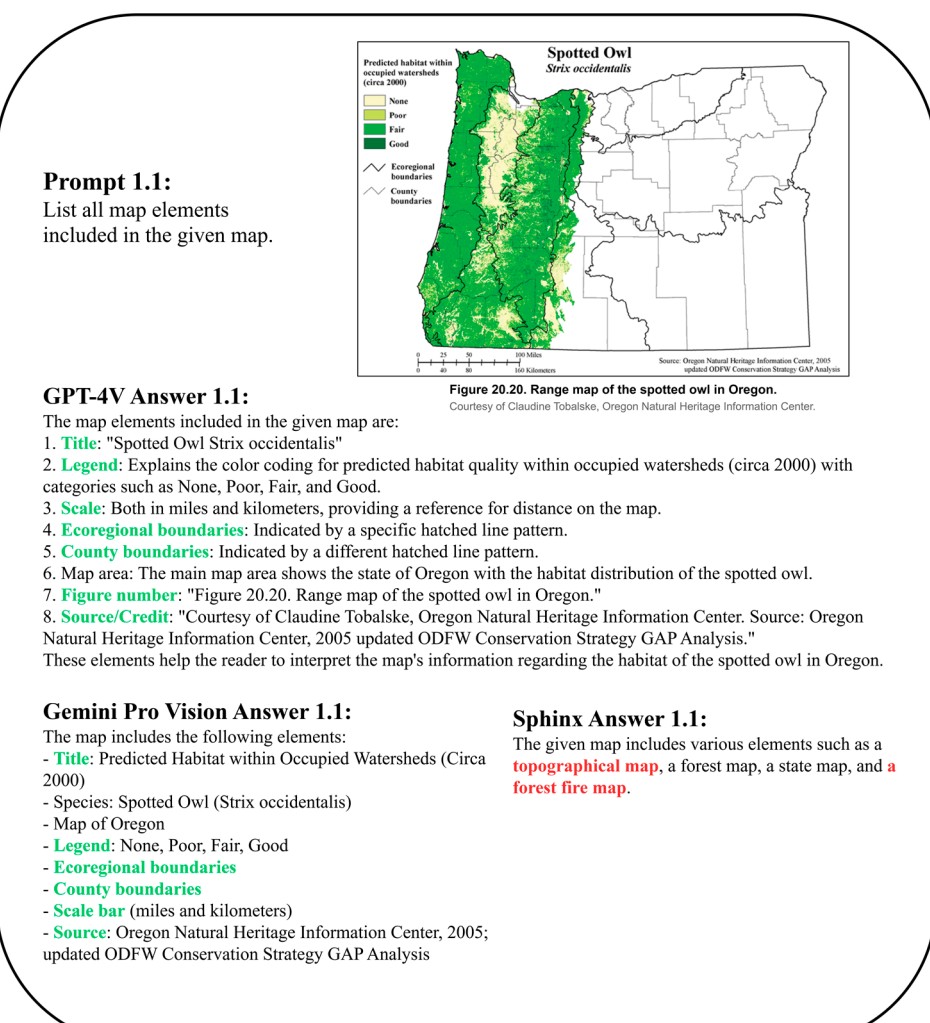

**Figure 2.** Prompt 1.1 and answers from three LMMs regarding to the map of spotted owls and its predicted habitats in Oregon, retrieved from [24], with proper answers highlighted in green, and incorrect answers highlighted in red.

In Answer 1.1, GPT-4V explicitly described all the map elements that appeared on the map. Gemini Pro Vision also interpreted the figure correctly but with less information, such as missing the figure number and author name in the figure caption. Sphinx only described the map but gave a false interpretation of the figure. By checking the details of the answers, the descriptions from GPT-4V and Gemini Pro Vision were mostly accurate and

correct, indicating their capabilities in extracting textual information from a map. Moreover, GPT-4V can link different elements to its use on the map. For example, it can first detect scales in two measurements and then explain that both scales (km and mile) provide a reference on the map. GPT-4V also interpreted the legend and explained how it works in layman's terms, while Gemini Pro Vision only listed the elements without explanation.

Furthermore, considering the possibility that existing maps from public sources were used for training the LMMs, we performed an additional element recognition test using newly generated maps. One hundred thematic maps were generated using demographic data collected from the 2019 American Community Survey through Census API, which includes 20 US states, 3 types of symbology (i.e., choropleth maps, graduated symbol maps, and dot density/distribution maps), and 3 different themes (i.e., population, unemployment rate, and per capita income). Notably, due to the limited capacity of Python packages in cartography, the maps used graticules/latitude and longitude grids to represent scales and north. Details of the thematic maps created can be found in the Supplementary Material.

Figure 3 shows an example of the prompt response from three LMMs. We evaluated the accuracy of each LMM's response to the question based on specified criteria, e.g., whether the response mentioned the legend. The results of this evaluation are summarized in Table 1. Among the models, GPT-4V demonstrated superior performance, and delivered a proper response that was more detailed than the other two LMMs. Detailed information about the images tested and the responses provided is available in the data repository.

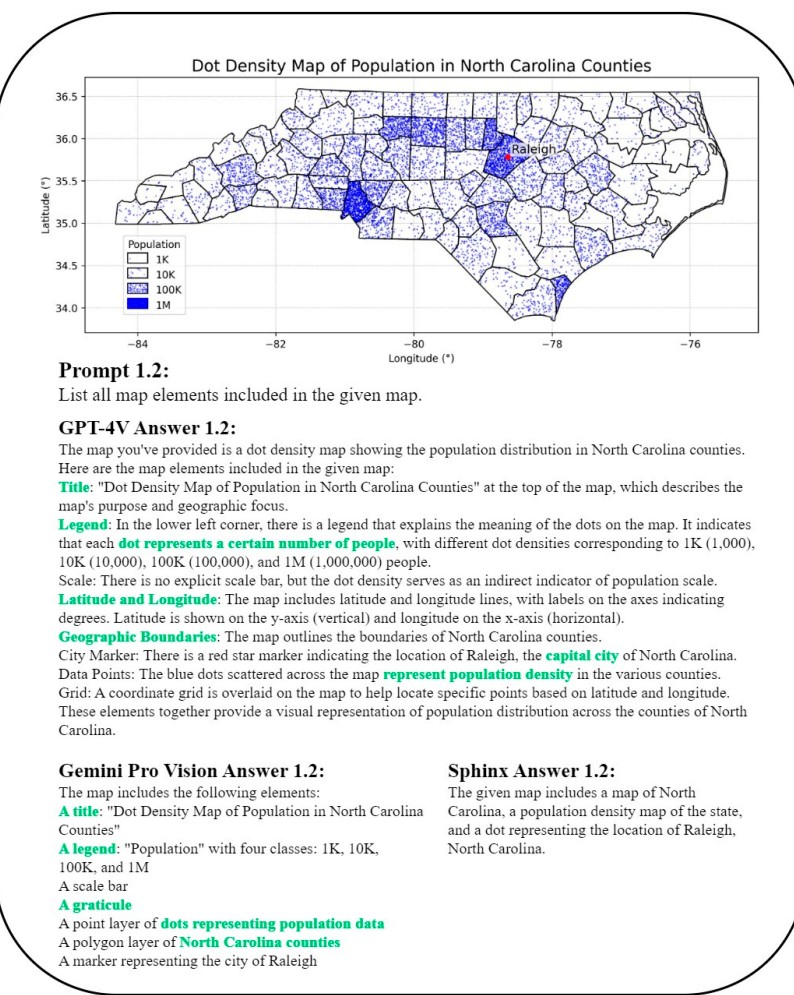

**Figure 3.** Prompt 1.2 and answers from three LMMs regarding to the generated maps, with proper answers highlighted in green.

**Table 1.** Performance of LMMs in the map element recognition task; the total number of maps tested in each LLM is 100.

| LMM | Answer Correctly | Mention Title | Mention Legend | Mention Symbol | Mention Boundary | Mention Capital | Mention Graticule |
|---|---|---|---|---|---|---|---|
| GPT-4V | 95/100 | 100/100 | 100/100 | 97/100 | 77/100 | 32/100 | 100/100 |
| Gemini Pro Vision | 14/100 | 99/100 | 100/100 | 26/100 | 32/100 | 25/100 | 85/100 |
| Sphinx | 0/100 | 0/100 | 13/100 | 1/100 | 1/100 | 1/100 | 0/100 |

## 2.2. Thematic Map Recognition

We further tested the ability of LMMs in reading different types of thematic maps. The first example tested in this prompt was retrieved from the 'Geographic Information Science & Technology Body of Knowledge' [27]. Four different types of maps are included in Figure 4, which are (1) dot density/distribution map, (2) proportional symbol map, (3) choropleth/graduated colored map, and (4) isoline/isarithmic map. We purposely hid the titles on the four maps, in order to concentrate on evaluating the image-reading capability rather than the text-recognizing capability of LMMs. In Answer 1.3, GPT-4V and Gemini Pro Vision correctly answered the questions, and GPT-4V further elaborated on different types of map representations and mentioned all four types of thematic maps in the example given. Furthermore, GPT-4V can explain how different types of thematic maps are related to the given axis labels, while Sphinx also tried to explain the relationship between axis and quadrants but failed. In this example, the responses by GPT-4V and Gemini Pro Vision can sufficiently meet our criteria for differentiating among the thematic map types, while the answer given by Sphinx lacked informative value.

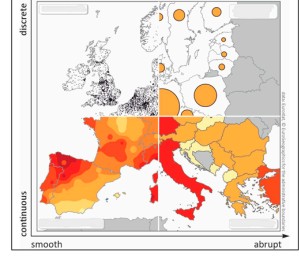

**Prompt 1.3:**
Vertical Axis and Horizontal Axis illustrates different types of map representation. List what types of thematic maps are in each quadrant of the figure attached.

**GPT-4V Answer 1.3:**
The figure attached appears to display a categorization of different types of thematic maps based on two axes: the vertical axis (from discrete to continuous) describes the nature of the variable, while the horizontal axis (from smooth to abrupt) describes the manner of change or distribution of the variable across space.
Quadrants from top-left clockwise are as follows:
1. Upper-left quadrant: This shows discrete and smooth thematic mapping. An example would be a **dot density map** where individual dots represent a certain quantity of a phenomenon, such as population, and are distributed evenly across the area to give a general sense of density or occurrence.
2. Upper-right quadrant: This illustrates discrete and abrupt thematic mapping, typically symbolized with **proportional symbols**. Here, discrete data (such as the number of certain facilities or events) is represented with symbols (like circles) whose size is proportional to the data value. These symbols are placed on the map where the data points exist.
3. Lower-right quadrant: This quadrant shows a continuous and abrupt type of thematic map. An example can be a **choropleth map** where data is aggregated by predefined areas (like countries or states) and each area is shaded or colored to represent a data range or category. The change between the categories is abrupt as it happens at the boundaries of the areas.
4. Lower-left quadrant: Here, continuous and smooth thematic mapping is presented, such as a **heat map** or an **isarithmic map**. This type of map visualizes continuous data (like temperature or elevation) as a smooth gradient, often using colors to represent different values, and the change between values happens gradually rather than at hard boundaries.

**Gemini Pro Vision Answer 1.3:**
Quadrant 1: Discrete-Smooth
 - **Choropleth map**
Quadrant 2: Discrete-Abrupt
 - **Graduated symbol map**
Quadrant 3: Continuous-Smooth
 - **Isarithmic map**
Quadrant 4: Continuous-Abrupt
 - **Area class map**

**Sphinx Answer 1.3:**
The figure attached contains four quadrants, each representing a different type of thematic map. The first quadrant displays a **vertical axis map**, which is commonly used to represent data or information in a vertical direction, such as population density or elevation. The second quadrant contains a **horizontal axis map**, which is used to represent data or information in a horizontal direction, such as temperature or precipitation. The third quadrant features a **vertical axis map** with a horizontal axis, which combines both vertical and horizontal data representation. The fourth quadrant contains a **horizontal axis map** with a vertical axis, which also combines both vertical and horizontal data representation.

**Figure 4.** Prompt 1.2 and Answers from LMMs regarding to the map of four thematic maps (names are redacted), retrieved from [27], with proper answers highlighted in green, answers that may be considered true under certain conditions highlighted in orange, and incorrect answers highlighted in red.

In addition, we tested the ability of LMMs to recognize different types of thematic maps using the same image set in Prompt 1.2 with the deliberate omission of their titles to focus purely on image interpretation capabilities. The results are summarized in Table 2 and reveal that GPT-4V accurately identified the symbology of 98 images, outperforming Gemini Pro Vision and Sphinx, which recognized 70 and 1 images, respectively. Based on the results above, GPT-4V exhibits a significant advantage in map reading, demonstrating exceptional performance that aligns with observations from other studies [32,33].

**Table 2.** Performance of LMMs in thematic map recognition tasks by types of maps tested (the rate of correct thematic map recognition indicated in the table).

| LMM | Choropleth Maps | Proportional Symbol Map | Dot Density Map |
|---|---|---|---|
| GPT-4V | 60/60 | 19/20 | 19/20 |
| Gemini Pro Vision | 60/60 | 1/20 | 9/20 |
| Sphinx | 1/60 | 0/20 | 0/20 |

As GPT-4V showed outstanding performances in the map reading tasks, we further explored other auxiliary tests for its map reading capabilities. Specifically, we tested map projection recognition (Prompt S1), map comparison with different scales (Prompt S2), domain-specific map reading (climate maps with Köppen climate classification in Prompt S3 and Local Indicators of Spatial Association in Prompt S4), optical illusion (Prompt S5), and high-resolution map reading with overwhelming information (Prompt S6). Due to the page limit, the prompts and results are attached in the Supplementary Material (Figures S2–S7). Compared with traditional map reading with human eyes, GPT-enabled map reading can save labor and time and provide an easier way to read large amounts of maps within a short time period.

## 3. Map Analysis

Map analysis involves precise measurements and the examination of spatial patterns from maps. Thematic maps are commonly used to discern spatial patterns. For instance, a graduated color map can be used to analyze precipitation variation across the contiguous United States (e.g., map image used in Figure 1), which reveals non-random, regional high and low clusters. Map analysis extends to examining spatial correspondences and differences in patterns between maps, revealing the complexity and insight of maps. In this section, four types of map analysis were used to evaluate the performance of GPT-4V, including (1) point pattern recognition, (2) bivariate pattern analysis, (3) visual detection of changes, (4) time-series analysis, and (5) comparison between different spatial scales.

### 3.1. Point Pattern Recognition
3.1.1. Point Pattern Analysis

Point pattern is one of the most common topics in map analysis. Point pattern analysis focuses on the spatial distribution of point data, usually including three types: clustered pattern, dispersed pattern, and random pattern. In this section, GPT-4V was tested for recognizing different types of point patterns. Figure 5, which indicates three types of point distribution, was retrieved from the book '*Mapping, Society, and Technology*' [29]. Points in blue circles indicate clustered distribution, points in yellow circles indicate random distribution, and points in red circles represent dispersed distribution. In Prompt 2.2 (Figure 6), we directly test GPT-4V's ability to distinguish between three types of point distribution. It turns out that point patterns in red and yellow circles cannot be recognized by GPT-4V. Considering that the map did not give any additional information on the colors of each circle, we conducted prompt engineering and added additional information explaining that each color of the circle represents a specific point distribution pattern. The response to our revised prompt (Prompt 2.2) was improved and could accurately point out each type of distribution. The step of adding additional information is also recommended in

the official prompt engineering guide [31]. Thus, when given enough information, GPT-4V could recognize different types of point patterns and can be used to expedite data analysis.

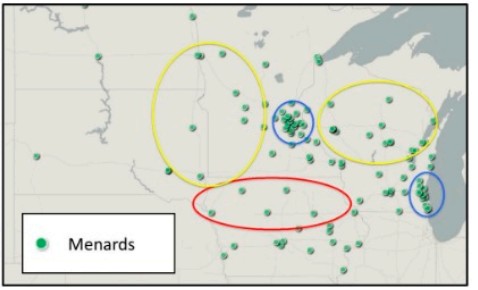

**Prompt 2.1:**
List the most likely point distribution (choose from random, dispersed, or clustered) in each circled area in the following map.

**GPT-4V Answer 2.1:**
- In the yellow circle on the left, the points are **dispersed**.
- In the red circle at the bottom, the points have a **random distribution**.
- In the yellow circle on the right, the points are **dispersed**.
- In the blue circles (both on the right and within the larger yellow circle), the points are **clustered**.

**Figure 5.** Prompt 2.1 and GPT-4V's Answer regarding to the map of hardware store clusters in the Midwest of the United States, retrieved from [29], color scheme used in this figure corresponds to the one described in Figure 4, indicating accuracy levels.

**Prompt 2.2:**
Each color represents one point distribution type. List the most likely point distribution (choose from random, dispersed, or clustered) in each circled area in the following map.

Same Figure Input as
**Prompt 2.1**

**GPT-4V Answer 2.2:**
Based on the provided map:

- The circled area in blue has a "**clustered**" distribution, as the points are grouped together quite closely in that region.
- The circled area in yellow on the right has a "**random**" distribution, with the points appearing to be scattered in no particular pattern.
- The circled area in red seems to have a "**dispersed**" distribution, with the points spread out more evenly and farther apart from one another compared to the clustered pattern.

It's important to note that these are the most likely point distribution types given the circled areas, but without additional context, the determinations are based solely on visual inspection.

**Figure 6.** Prompt 2.2 and GPT-4V's Answer after prompt engineering (adding additional information) on Prompt 2.1, with proper answers highlighted in green.

### 3.1.2. Bivariate Point Pattern Analysis

In addition to simple point pattern analysis, we also tested if GPT-4V can recognize different types of bivariate point patterns. Similar to previous prompts, we tested three types of bivariate point patterns, namely, clustered, regular, and random (Figure 7), using the figure from a relevant publication [34]. GPT-4V's answer can correctly point out the most likely pattern for each point distribution. Additionally, GPT-4V uses its extensive language model to explain the characteristics of clustered, regular, and random distributions. Then, we used the crime and income data in Chicago as the case study (Figure 8). We collected point data from the crime dataset in Chicago and collected income data in each census block group. Then, we overlaid the selected two crime types (burglary and theft) on a graduated color map of income and tested if GPT-4V can recognize such a complex map. To our surprise, GPT-4V not only accurately described that the distribution of theft and burglary over the map is more concentrated in certain areas, but also gave a guess that theft data may be more related to income based on visual inspection. We later examined

the result given by GPT-4V by calculating the Pearson correlation coefficient and *p*-value between the number of burglary/theft and income at the census block group level, and the result showed that GPT-4V's guess was correct in that both correlations are significant, and theft has a stronger correlation with income than burglary (see Table S1).

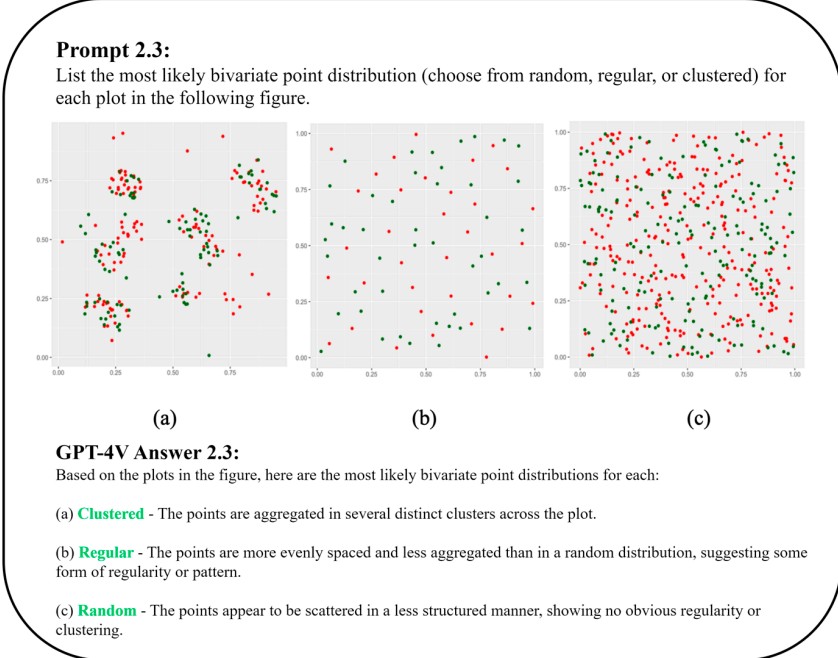

**Figure 7.** Prompt 2.3 and GPT-4V's Answer regarding the map of bivariate point distributions (represented as green and red dots), retrieved from [34], with proper answers highlighted in green.

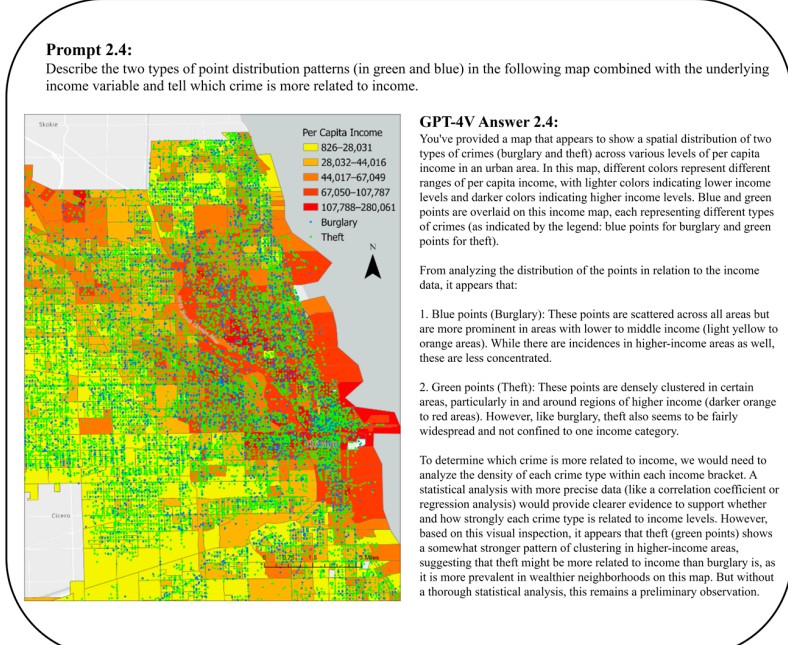

**Figure 8.** Prompt 2.4 and GPT-4V's Answer regarding the map of bivariate point distribution (burglary and theft) overlaid with income background layer, with crime data collected from https://data.cityofchicago.org/Public-Safety/Crimes-2022/9hwr-2zxp/data, and income data collected from American Community Survey 2021 5-Year Estimates, both of which were accessed on 30 December 2023.

## 3.2. Comparison between Maps

The comparative analysis of maps is a fundamental aspect of cartographic evaluation, going beyond mere visual inspection to understand the underlying patterns, changes, and inconsistencies between different map representations. This process is critical in discerning how various mapping techniques, data sources, and temporal changes impact the portrayal of geographic information. For instance, comparing topographic maps from different eras can reveal landscape changes, while juxtaposing thematic maps derived from different classification algorithms might highlight variations in environmental monitoring. In this section, the map comparison is organized into three distinct yet interconnected categories: Firstly, visual detection of changes in maps, focusing on the temporal scale with a pair of maps ($n = 2$), to identify and analyze alterations over time. Secondly, time-series analysis, which extends the temporal scale to multiple maps ($n > 2$), allows for a more dynamic and longitudinal understanding of changes. Lastly, the comparison across different spatial scales examines how geographic information is represented and interpreted differently depending on the areal unit used in the map. Each category leverages GPT-4V's advanced capabilities to offer insightful and comprehensive map comparisons, enhancing the understanding of geographical data and their implications.

### 3.2.1. Visual Detection of Changes in Maps

We first start our map analysis and comparison test with the visual detection of changes in maps. Prompt 2.5 is based on the high-resolution nighttime light (NTL) images (30 m) in the Houston area before and during the 2021 Winter Storm Uri (in Figure 9). We georeferenced the NTL images from NASA with labels provided by ArcGIS. The answer by GPT-4V demonstrates the capability to accurately identify areas experiencing power outages. GPT-4V can not only deduce from the information presented in the maps which locations are more or less affected by power outages, e.g., southwest near Missouri City experienced a decrease in NTL intensity, but can also infer the power outage status of certain areas that are not explicitly marked on the map, e.g., northwest near Cypress experienced an NTL intensity reduction. GPT-4V also integrates its large training data to assess the outage situation in specific areas, e.g., industrial areas near Baytown and La Porte maintain consistent lighting. Moreover, GPT-4V offers concrete analytical suggestions and identifies several limitations of assessing power outages using NTL data in the answer, such as cloud cover, time of image capture, and sensor used.

### 3.2.2. Time-Series Analysis

Extending the temporal scale from two to more, the test of time-series analysis on GPT-4V is conducted to evaluate its ability for visual detection of changes. We used the annual precipitation maps generated from the interactive mapping platform, Climate at the Glance, under the Climate Monitoring product provided by NOAA National Center for Environmental Information (NCEI) from 2000 to 2020 at a 5-year interval (i.e., 2000, 2005, 2010, 2015, 2020). Five maps were included in the message to GPT API by attaching URLs to their cloud location in Google Drive (Figure 10). Prompt 2.6 shows that GPT-4V can accurately provide a comprehensive qualitative assessment of the maps provided. Specifically, GPT-4V can provide the visual detection of changes between multiple map sets within a short amount of time, which is superior to human beings when reading multiple maps. Moreover, its outstanding ability of textual content extracting gives an additional correction for GPT-4V to understand and correct its answer to the prompt, like giving detailed changes in numbers shown on maps. As GPT-4V explained in its answer, it can only provide qualitative assessment (highlighted in bold in Figure 10) and may still lack accuracy in quantities. In most cases, data behind maps are rarely given simultaneously, making GPT-4V's accurate qualitative assessment more valuable. However, without massive experiments for quality assurance and quality control, GPT-4V should be carefully applied when conducting a time-series analysis.

**Prompt 2.5:**

List the visible changes to determine which areas were most impacted in the following nighttime images of the Houston area before and during the 2021 Winter Storm Uri.

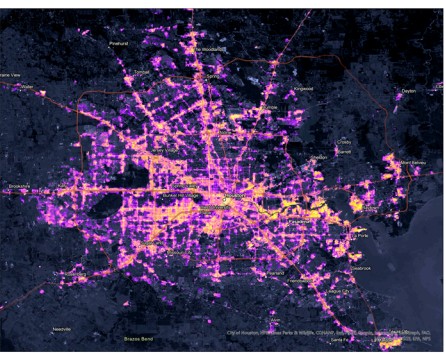 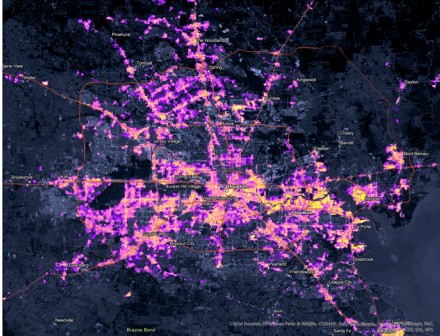

**GPT-4V Answer 2.5:**

To analyze these nighttime images effectively, we should look for changes in the intensity and distribution of the lights to determine the impact of the winter storm on different areas within the Houston region. Specifically, we are looking for areas where lights have diminished or are no longer visible, which could indicate power outages or reduced human activity.

Here are observable differences between the two images that suggest areas most impacted by the winter storm:

1. Some areas, particularly on the outskirts and less dense regions around Houston, show less intense light or complete loss of light. This may indicate power losses in these areas.

2. Within the city's core and densely populated areas, there appears to be a consistent intensity of light, although some small, localized reductions can be noticed.

3. Suburban areas around the city, such as to the northwest near **Cypress** or to the southwest near Missouri City, show patches where the intensity of light has visibly decreased.

4. Some major **roadways** that are brightly lit in the first image seem dimmer in the second, which might be related to reduced traffic or street light outages.

5. **Industrial** areas, such as those to the east near Baytown and La Porte, appear to maintain consistent lighting, possibly indicating that critical infrastructure there continued to operate.

6. While **residential** and **commercial** areas showcase some variations, the differences are not uniform across the board, suggesting that the storm's impact was varied and possibly related to the local infrastructure's robustness or emergency preparedness measures.

To provide a more definitive analysis, one would typically cross-reference these observations with additional data such as power outage reports, emergency service data, and infrastructure resilience assessments.

Please note, however, that while visual inspection can indicate relative changes in power usage and outages, for a precise and detailed analysis, ground truth information and additional data sources would be necessary. Factors such as cloud cover, time of the image capture, and the exact nature of the sensors used to create the images can all influence the interpretation.

**Figure 9.** Prompt 2.5 and GPT-4V's Answer regarding the map comparison between two NTL images in Houston on 7 February 2021 (before the winter storm) and 16 February 2021 (during the winter storm), NTL images retrieved from NASA (https://appliedsciences.nasa.gov/our-impact/news/extreme-winter-weather-causes-us-blackouts, accessed on 30 December 2023), additional insights identified by GPT-4V are highlighted in bold.

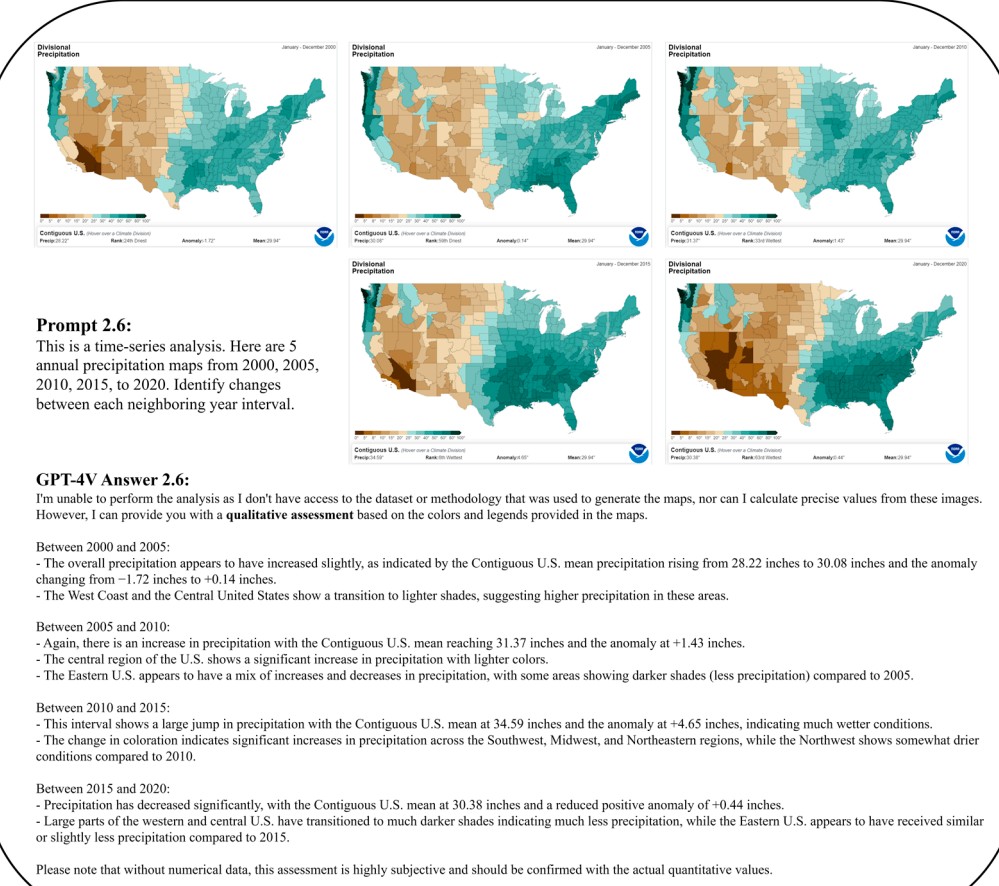

**Figure 10.** Prompt 2.6 and GPT-4V's Answer regarding the map for time-series analysis using divisional precipitation data from 2000 to 2020 at a 5-year interval (i.e., 2000, 2005, 2010, 2015, 2020), retrieved from the interactive mapping platform, Climate at the Glance, under the Climate Monitoring product provided by NOAA NCEI (https://www.ncei.noaa.gov/access/monitoring/climate-at-a-glance/divisional/mapping, accessed on 30 December 2023).

### 3.2.3. Comparison across Different Spatial Scales

Spatial scale is important in spatial analysis, and comparison across different spatial scales can not only show a detailed view of spatial aggregation, but also address the modifiable areal unit problem (MAUP), one of the popular and important questions in geography, which is a statistical bias that can significantly affect the interpretation of data in geospatial analysis. Here, we tested GPT-4V's ability to distinguish spatial patterns across different spatial scales in the same temporal scale. Similarly, maps in three spatial scales, i.e., state, divisional, and county, generated from Climate at the Glance under the Climate Monitoring product provided by NOAA NCEI, were collected and applied in Prompt 2.7 (Figure 11). GPT-4V's answer illustrates that it can tell the differences in scales when comparing three maps, and the differences lead to a variation in resolution whereby maps with a smaller scale have a higher resolution. Specifically, it mentioned localized patterns in variability, which is a key component in the map comparison across three spatial scales. Thus, we followed up with Prompt 2.8 (Figure 12) to ask GPT-4V to identify which areas have the largest variability. The answer first provides a workflow on how to observe areas with the largest variability. Then, it gives some potential candidate states, like Texas, California, and Midwestern states, but it does not firmly confirm its observation on areas with the largest variability. The answer additionally introduced the concept of microclimates, which was absent from our original prompt, indicating that GPT-4V may use its pre-trained large language model to improve its map reading and analysis results.

By following up with specific questions, GPT-4V can provide a relatively vague answer and phrases its answer with high uncertainties, like using "seems", "relatively", "might not", etc. Overall, GPT-4V could distinguish differences across spatial scales, but its assessment on maps stays preliminary and qualitative.

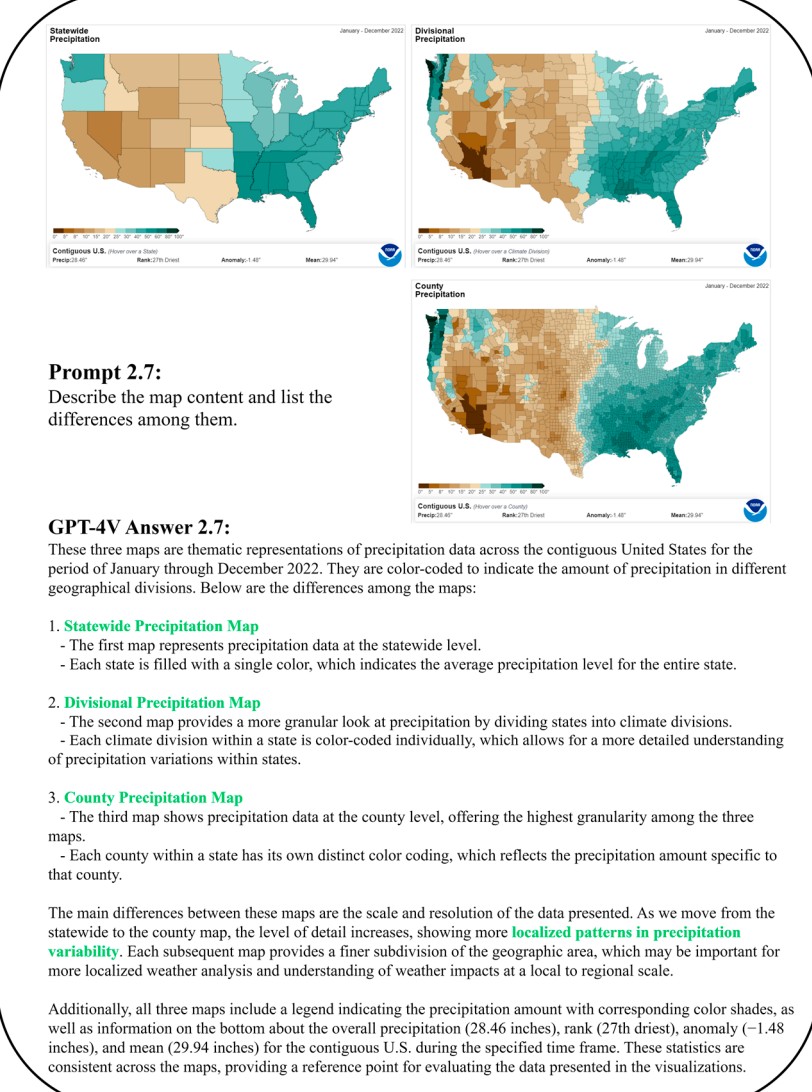

**Figure 11.** Prompt 2.7 and GPT-4V's Answer regarding the map comparison across three spatial scales (statewide, divisional, and county) using annual precipitation data in 2022, retrieved from the interactive mapping platform, Climate at the Glance, under the Climate Monitoring product provided by NOAA NCEI (https://www.ncei.noaa.gov/access/monitoring/climate-at-a-glance/divisional/mapping, accessed on 30 December 2023), with proper answers highlighted in green.

**Prompt 2.8:**
(Follow-up Prompt to **Prompt 2.7**) Identify places that have the largest variation across the three spatial scales (State, Divisional, and County).

| Same Figure Inputs as **Prompt 2.7** |

**GPT-4V Answer 2.8:**
When comparing the three maps of precipitation for the contiguous United States, the variation in the representation of precipitation data reveals itself more clearly in areas where the coloring changes significantly from the statewide level to the county level.

Areas of larger variation can be spotted by looking for:

- States that display a mostly uniform color on the statewide map but break down into a much more varied pattern at the divisional or county level.
- Counties within the same state that show a markedly different color compared to neighboring counties or compared to the average color of the state on the statewide map.

Here are some places that seem to have larger variations:

1. **Western States**:
   - For instance, **California** shows a gradient from darker to lighter shades indicating **more variability in precipitation**, especially when considering the county-level map where certain counties reveal significantly different precipitation levels.

2. **Central States**:
   - States like **Texas** also have **significant variability** when moving from the divisional to county level. Some areas in Texas that appear relatively uniform at the divisional level are divided into many varying shades at the county level.

3. **Eastern States**:
   - Similarly, the southeastern states such as **Florida**, **Georgia**, and **the Carolinas** show relatively **uniform precipitation** at the state level, but as we move into the more detailed maps, variations in individual counties become more apparent.

4. **Midwestern States**:
   - States in the **Midwest** exhibit **significant differentiation** at the county level as well. Here, we observe a distinct distribution of precipitation in numerous counties which isn't as apparent at the state or divisional scale.

These observations indicate that while some regions have a relatively homogenous climate, others exhibit diverse microclimates that result in varied precipitation patterns. The county-level data is crucial for understanding these localized climates, as it provides the highest resolution of data, highlighting geographical and topographical influences on precipitation that might not be visible at broader scales.

**Figure 12.** Prompt 2.8 and GPT-4V's Answer following up the prompt response in Prompt 2.7, insights for different locations were highlighted in bold.

## 4. Discussion

This study evaluated the latest version of GPT for vision, GPT-4V, across various facets of map reading and analysis and compared the map reading ability of GPT-4V with other LMMs. The findings demonstrate that GPT-4V is not only proficient in executing fundamental map reading and analysis tasks but is also skillful at identifying complex spatial patterns. Based on our results, we have summarized several advantages and disadvantages of GPT-enabled map reading and analysis in Table 3.

**Table 3.** Pros and cons of GPT-4V's application in map reading and analysis with supporting prompts.

|  | Pros/Cons | Map Reading | Map Analysis |
|---|---|---|---|
| Pros | 1. Accurate info retrieval | All Prompts | All Prompts |
|  | 2. Geographic knowledge | Prompt 1.3, S3 | Prompt 2.5 |
|  | 3. Comprehending complex symbology | Prompt 1.3, S6 | Prompt 2.4, 2.6 |
|  | 4. Spatial pattern recognition | Prompt 1.2 | All Prompts |
|  | 5. Picking up details | Prompt S6 | Prompt 2.4, 2.5 |
|  | 6. Understanding domain-specific maps | Prompt 1.2, S3, S4 | Prompt 2.4 |
|  | 7. Efficiency | All Prompts | All Prompts |
| Cons | 1. Constraints in precision | Prompt 1.2 | Prompt 2.4, 2.6 |
|  | 2. Dependence on prompt engineering | N/A | Prompt 2.1, 2.2 |
|  | 3. Difficult results validation | All Prompts | All Prompts |
|  | 4. Limited explicability | All Prompts | All Prompts |
|  | 5. Reproductivity concern | All Prompts | All Prompts |
|  | 6. Refusal behavior | All Prompts | All Prompts |

*4.1. Advantages*

1. Accurate Information Retrieval

GPT-4V demonstrates a remarkable capability of information retrieval from maps, especially embedded textual information. Throughout all prompts in the map reading and analysis sections (Sections 2 and 3), GPT-4V consistently and successfully extracted the textual information, and provided a precise description of map elements, such as legend items, figure caption, and scales. This skill is particularly important when reading domain-specific maps with complicated scales and legend items, where precision is crucial.

2. Geographic Knowledge

GPT-4V can connect places observed from a map to its pre-trained geographic knowledge. For instance, as demonstrated in Prompt 2.5, GPT-4V was able to extend beyond the map's displayed content and reference geographical information about locations not shown in the image, such as Cypress near the Houston area. Similarly, in Prompt S2, the geographical relationship between Hollywood and North Miami Beach was not explicitly mentioned on the map, yet GPT-4V was able to acquire position information about Hollywood and North Miami Beach. Although there was an error in our experimental results (the correct statement should be that Hollywood is north of North Miami Beach), such a capability in map reading is rare and noteworthy. This suggests that GPT-4V's training model encompasses the geographic location information of these places, and that it understands how such information is interrelated. Consequently, GPT-4V is able to establish spatial connections, which proves to be incredibly useful in the context of map reading and analysis. By drawing from its large background information, GPT-4V can relate to and incorporate external geographic information, thereby enhancing the depth and accuracy of its interpretations within the map's framework.

3. Comprehending Complex Symbology

GPT-4V's advanced capabilities in map reading, particularly in understanding complex map symbology (e.g., thematic maps with different symbology types, color schemes, and classifications), give it a significant advantage over traditional methods. Complex legends can often confuse human readers, but GPT-4V, supported by a large pre-trained language model, consistently identifies similar or analogous information to use as references. This allows for a more accurate interpretation of the information in the legends. For instance, as seen in Prompts 1.3 and 2.4, GPT-4V was able to discern color information corresponding to different grades or symbols representing various data representations. GPT-4V goes beyond merely extracting this information; its real value lies in utilizing the large language model to comprehend and elucidate the significance of this information. In an example like Prompt S6, GPT-4V did not just list the content indicated by the legend but also inferred conclusions about the data, such as associating darker shades with higher income levels.

4. Spatial Patterns Recognition

GPT-4V also exhibits an outstanding capacity for pattern recognition, as evidenced by its performance in experiments involving point patterns. In scenarios such as those presented in Prompts 2.1 and 2.2, GPT-4V successfully identified different point patterns—dispersed, regular, and random—matching the discernment capabilities of the human eye. Moreover, in tasks related to comparison, GPT-4V proved its mettle, as seen in Prompt 2.5, where it accurately detected changes in nighttime light before and during the winter storm. While there were occasional instances where it could not provide image comparison responses, generally, when provided with ample contextual conditions and after some prompt engineering, GPT-4V could effectively recognize and compare images.

5. Picking Up Details

GPT-4V shows an exceptional ability to process maps with high-resolution and precise information, outperforming human capabilities in certain aspects. For example, in Prompt S6, where maps contain an overwhelming number of elements, humans may struggle to

quickly locate the necessary information amidst the complexity. GPT-4V, on the other hand, can identify textual information promptly with a wide range of associated background knowledge. This capability presents a significant opportunity, particularly considering the vast archives of historical maps that remain unused due to their complexity. The widespread application of GPT-4V to such historical maps could extract and weave together a wealth of information, creating a new network of interconnected map information. This network has the potential to inspire novel findings, transforming the way we comprehend historical cartography and its narratives.

6. Understanding Domain-Specific Maps

GPT-4V has the potential to greatly assist laymen in reading domain-specific maps, which often come with a steep learning curve due to map complexity. Maps rich in specialized content, like Local Indicators of Spatial Association (LISA) maps (Prompt S4) or those employing the Köppen climate classification (Prompt S3), typically require a solid background in the subject matter to be fully understood. Prior to GPT-4V, a person would need to turn to search engines to supplement their understanding with background knowledge. GPT-4V, however, can streamline this process by directly providing relevant information, paving a new pathway for understanding domain-specific maps. By leveraging its pre-trained large language model, it can offer comprehensive and related information, aiding people from non-specialized fields in quickly grasping the content of complex maps. This feature of GPT-4V not only enhances the accessibility of specialized geographical data but also enriches the user's learning experience by simplifying the acquisition of domain knowledge.

7. Efficiency

GPT-4V improves the efficiency of reading and analyzing maps when compared with humans. In most of our tests, the response time from OpenAI API is less than 20 s, which is notably faster than what it typically takes for a person to observe a map and type the descriptions, e.g., ranging from 30 to 95 words per minute (wpm) with a mean around 50 wpm [35]. A sample response in our test is around 200 words, which may take 4 min to type, and the GPT-4V's response time is approximately 12 times faster than that of a human, while this calculation does not even factor in the additional time humans require for organizing and structuring their responses. Furthermore, in scenarios that require reading or analyzing multiple maps, the time saving becomes even more pronounced. Humans may require substantially more time to comprehend each map individually, whereas GPT-4V can be set to process multiple requests simultaneously through the GPT-4 API.

*4.2. Disadvantages*

1. Constraints in precision

In the evaluation of GPT-4V's performance, a notable concern is its accuracy in quantitative assessments. During tests, such as those associated with Prompt S2, GPT-4V demonstrated limitations in providing precise scale measurements from two maps. In response to Prompt 2.6, GPT-4V excels in qualitative analysis, suggesting its suitability for descriptive tasks rather than accurate quantitative evaluations. GPT-4V consistently indicated that it does not support quantitative analysis on maps, which implies that GPT-4V can effectively interpret and describe map content but remains restricted for quantitative tasks. Users should be aware of this limitation and may prefer to utilize GPT-4V in contexts where descriptive insight and qualitative interpretation are the primary objectives.

2. Dependence on Prompt Engineering

The application of GPT-4V in map analysis often requires careful prompt engineering. In most cases, initial inquiries rarely yielded comprehensive answers (e.g., Prompt 2.1). Thus, refining the prompts is essential to guide the AI toward the desired range of responses, enhancing the relevance and accuracy of the information provided. In other words, prompt engineering is a cornerstone of effective AI deployment. This iterative approach of fine-

tuning prompts is a critical step to effectively leverage GPT-4V's capabilities in map reading and analysis, necessitating significant effort and collaboration as an additional step.

3. Difficult Results Validation

The current version of GPT-4V encounters technical difficulties, particularly with analyzing graphs or text involving varying colors or styles, such as solid, dashed, or dotted lines. This challenge might be caused by the way data are parsed and fed into the GPT model. While immediate improvements on this front may be challenging, future updates and releases from OpenAI could potentially address and alleviate these issues.

4. Validation Concern

The validation of results provided by GPT-4V presents a concern. Currently, the validation of its experimental outcomes is conducted by the authors, which could potentially introduce biases. Consequently, our experiments do not definitively conclude that GPT-4V always delivers high-quality results. A solution in future studies could be to increase the number of experiments to further explore the stability and reliability of GPT-4V.

5. Limited Explicability

GPT-4V is like a "black box", with its underlying principles and logic challenging to explain. In essence, it relies on its large pre-trained language model and deep neural networks in transformers to generate responses. The architecture of GPT-4V is conceptually straightforward, yet providing a clear explanation of its internal process is complex. The responses it generates can vary, sometimes depending more on the information extracted directly from the map, and at other times, leaning heavily on its pre-trained data. The balance between extracting visual content from the map and utilizing information from its large language model training is not easily controllable. This variation in results further complicates the explanation of GPT-4V's mechanism.

6. Reproducibility Concern

The reproducibility and performance of GPT-4V in our experiments raise concerns. Due to the nature of large language models and GPT's characteristics, consistent results cannot be expected. Thus, there is a possibility that our current findings could be coincidental. Moreover, the scope of our experiments may be limited, indicating GPT-4V's capabilities only under specific conditions. Its performance in more complex tasks remains uncertain and requires further experiments. Although GPT-4V uses complicated algorithms to predict the next word, which may impact reproducibility, it is likely to produce similar outcomes when given the same prompts and images. However, there is still a need for more experiments to help better understand and validate GPT-4V's capabilities in map reading and interpretation.

7. Refusal Behavior

It is noteworthy that in our experiment, there were several instances where GPT-4V refused to respond to our map-related queries. As outlined in the GPT-4V system card, such non-responsiveness can be triggered by issues like harmful content, privacy concerns, cybersecurity, or multimodal jailbreaks [22]. Yet, in our case, these triggers did not evidently apply to our prompts. This discrepancy suggests a possible misidentification of such triggers by the system. While revising the phrasing of our prompts sometimes resulted in an effective response from GPT-4V, this inconsistency highlights a need for clearer guidelines in GPT-4V's future documentation, particularly regarding what types of prompts the system can process.

*4.3. Recommendations*

Based on the aforementioned advantages and disadvantages, we produce the following recommendations for when and how to best use GPT-4V for map reading and analysis:

First, GPT-4V can be a great assistant with fast information retrieval from large-volume, high-frequency, high-resolution maps with complex symbology. Our experiments have

demonstrated its capabilities of correctly reading maps of various types, styles, and topics. Thanks to its machine nature that allows for programming and automation, it is safe to recommend using GPT-4V for processing batch maps for reading and analysis simultaneously. This multi-processing capability is particularly valuable in fields that require processing a large volume of map data in a short time, such as real-time monitoring of hotspots of crime incidents, car accidents, or disease outbreaks. Similarly, it is beneficial for long-term large-scale map-based monitoring, such as coastal erosion detection from time-series satellite-image maps, or drought monitoring from daily precipitation maps. Automating the process of examining batches of maps and summarizing observable patterns in writing using GPT API is practical low-hanging fruit benefiting from this new technology.

Second, it significantly lowers the learning curve of map reading for many. Proper map reading requires basic knowledge of cartography and geography, and it takes practice. Even for someone who is skillful and experienced, it is still challenging to read reference maps of unfamiliar geographic regions, or thematic maps of unfamiliar topics. GPT-4V can well address such challenges thanks to its pre-trained geography and domain knowledge. For instance, GPT-4V serves as a "local guide" to explain map labels of unfamiliar places or place names in a foreign language. It can also translate jargon that appears in maps to layman's terms to ease the way of understanding spatial patterns.

Third, though GPT-4V showed spectacular performance in most tasks tested, it still presents some limitations in recognizing patterns in maps (Figure S8). Such mistakes do not always occur, making it hard to identify the core issue. Thus, our recommendation for the use of GPT-4V in reading and analyzing maps is to run it more than once and synthesize the results derived from responses to avoid using casual false interpretations from the model.

Fourth, GPT-4V can facilitate the research process in geographic information science. Spatial pattern recognition from maps often serves as the first step of exploratory spatial data analysis, especially in the big data era. Various research hypotheses can be formed based on the observed patterns. And these hypotheses can be further validated with confirmatory analysis using real-world data to form new empirical findings or even new theories. GPT-4V can significantly enhance this process by mining spatial patterns from maps, summarizing the patterns in writing, and comparing them with the literature in its pre-trained database to suggest which patterns are worth further study.

## 5. Conclusions

This study explores using the latest GPT-4V for map reading and analysis. Our experiments focused on map element recognition, thematic map recognition, and advanced map analysis including point pattern recognition and comparative analysis across different spatial and temporal scales, which demonstrate GPT-4V's capability to efficiently extract information from various maps and perform basic visual analytics. We also discussed and summarized its pros and cons in map reading and analysis. Its effectiveness in tasks, such as identifying changes in satellite images between and during a winter storm, suggests a promising application in automating map analysis. However, there is room for improvement in relation to its diminished accuracy in visual content extraction, need for prompt engineering, technical difficulties, validation concerns, "black box" nature, and issues with reproducibility and performance.

GPT-4V's strengths, like its ability to rapidly process high-resolution images and assist non-experts in understanding complex maps, are offset by its limitations in providing precise quantitative analysis and the need for carefully engineered prompts to yield accurate results. Moreover, technical challenges and the inherent complexity of its algorithm make it difficult to fully understand and predict its behavior. Despite these challenges, the future of GPT-4V in map reading and analysis is promising. With anticipated technological advancements and updates, many of the current limitations could be mitigated, potentially expanding the scope and effectiveness of GPT-4V in geospatial analysis.

The dynamic evolution of GPT-4V, influenced by both technological progress and policy changes, indicates a future where its capabilities could be significantly enhanced.

This study lays the groundwork for further exploration and application of GPT-4V and similar AI tools in the field of cartography and geospatial analysis, ushering in a new era of AI-enabled map interpretation. There exists a future possibility in data science in which the steps of making and reading maps may be bypassed. When AI tools advance to a certain level, they can be used to directly retrieve useful patterns and information from abundant geospatial data, without the necessity for designing maps, and then visual observations can be made.

The advantages and limitations discussed are specific to the GPT-4-vision-preview version and its training data as of September 2023. With anticipated updates, the highlighted advantages could be enhanced, and the current limitations might be mitigated. Conversely, future policy updates might impose restrictions on certain map interpretation functionalities that are presently available. This dynamic nature of AI development suggests that the effectiveness and scope of GPT-4V in map analysis will continue to evolve, influenced both by technological advancements and policy decisions.

There are several meaningful future directions to extend the current study of GPT-enabled map reading and analysis. First, it is worth testing reading maps in formats other than static images. All the experiments conducted in this study use maps in common digital image file formats such as JPEG. However, modern maps are presented in many other ways, including the animated maps in Graphics Interchange Format (GIF) or in videos, interactive maps in web browsers, and maps embedded in smartphone applications. Evaluating how well GPT-4V can read and comprehend maps in such other formats can likely enhance its applicability. Second, reading maps can be more inclusive. By leaving on the latest voice control feature of GPT, it is a natural next step to have it read maps for vision-impaired people. Such a special group of users can use voice commands to ask what a map shows and hear map descriptions instead of seeing them. Last, but not least, combining GPT-enabled map analysis with more advanced spatial analysis is a promising direction to enhance future research. For instance, a plausible approach is to transfer the spatial patterns inspected by GPT-4V from maps to advanced analytical or modeling toolkits for further validation. In that scenario, GPT-enabled map reading and analysis will be a critical step of the research pipeline, in which observing, analyzing, processing, thinking, reasoning, summarizing, and writing will all be highly automated with future AI tools.

**Supplementary Materials:** The following supporting information can be downloaded at: https://www.mdpi.com/article/10.3390/ijgi13040127/s1, Figure S1: Comparison between responses using different prompts (Prompt S1.1, S1.2, S1.3) after prompt engineering; Table S1. Pearson Correlation Coefficient and *p*-value between crime and income level in Chicago at block-group level; Figure S2: Prompt S2 and GPT-4V's Answer regarding to the map projection, image retrieved from The nature of geographic information (https://www.e-education.psu.edu/natureofgeoinfo/c2_p26.html, accessed on 30 December 2023) [36]; Figure S3: Prompt S3 and GPT-4V's Answer regarding to the map comparison with different spatial scales, image retrieved from Google Maps (https://www.google.com/maps, accessed on 30 December 2023); Figure S4: Prompt S4 and GPT-4V's Answer regarding to the reading of domain-specific maps, making Köppen climate classification system as an example, image retrieved from Wikipedia (https://upload.wikimedia.org/wikipedia/commons/thumb/1/1e/World_K%C3%B6ppen_Classification_(with_authors).svg/675px-World_K%C3%B6ppen_Classification_(with_authors).svg.png, accessed on 30 December 2023); Figure S5: Prompt S5 and GPT-4V's Answer regarding to the reading of domain-specific maps, making map of Local Indicators of Spatial Association (LISA) as an example, image retrieved from StackExchange (https://stats.stackexchange.com/questions/335919/how-to-interpret-lisa-clustering-maps, accessed on 30 December 2023); Figure S6: Prompt S6 and GPT-4V's Answer regarding to the optical illusion, image retrieved from Scientific American (https://www.scientificamerican.com/gallery/optical-illusion-by-land-or-by-sea/, accessed on 30 December 2023); Figure S7: Prompt S7 and GPT-4V's Answer regarding to the map information extraction from high-resolution image with overwhelming labels, image generated from data from American Community Survey 2021 5-Year Estimates in Texas at the county level; Figure S8: Mistakes in GPT-4V's answer in additional experimented prompts, highlighted in red.

**Author Contributions:** Conceptualization, Jinwen Xu and Ran Tao; methodology, Jinwen Xu and Ran Tao; validation, Jinwen Xu and Ran Tao; formal analysis, Jinwen Xu; investigation, Jinwen Xu; resources, Jinwen Xu and Ran Tao; data curation, Jinwen Xu; writing—original draft preparation, Jinwen Xu; writing—review and editing, Jinwen Xu and Ran Tao; visualization, Jinwen Xu; supervision, Ran Tao; project administration, Jinwen Xu and Ran Tao; funding acquisition, Ran Tao. All authors have read and agreed to the published version of the manuscript.

**Funding:** This research received no external funding.

**Data Availability Statement:** All the prompts, programming codes, and resulting maps in this study can be downloaded at: https://github.com/jinwenxu/GPT-4V_Mapping (acceded on 29 February 2024).

**Conflicts of Interest:** The authors declare no conflicts of interest.

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
