# Peer review of "Map Reading and Analysis with GPT-4V(ision)"

_ijgi, doi:10.3390/ijgi13040127_

Round 1

Reviewer 1 Report

Comments and Suggestions for Authors

The manuscript titled “Map Reading and Analysis with GPT-4V(ision)” discusses an interesting topic, namely, the use of LLMs for map interpretation. The manuscript as such is well-structured and the flow of text is good. The idea of the authors can easily be followed and the chosen examples for experimentation are good. However, the methodology and especially the evaluation needs to be significantly improved. While the use of examples with existing results for comparison are adequate to assess initial results, I do miss any evaluation concerning the overall performance of GPT-4V in terms of how it performs against other state-of-the-art LLMs/interpreters. The authors have shown that there are limitations and to see how far behind/ahead the selected LLM, i.e. GPT-4V, is, the authors either need to establish a baseline or – as mentioned before – perform the same experiment with multiple LLMs. Furthermore, while the discussion section is well thought and argued, I do miss concrete suggestions for improvement of currently identified difficulties and shortcomings.

Another point is the topic of prompt engineering, which plays a significant role in the application domain selected by the authors. They themselves have shown during their experiments that the way how the questions were framed can have a significant impact on the overall results. Hence, this aspect should be addressed more thoroughly within the manuscript and – at least – demonstrate some approaches and solutions in this regard for the provided cases.

Comments on the Quality of English Language

The quality of English language is fine.

Reviewer 2 Report

Comments and Suggestions for Authors

This paper verifies the ability of GPT-4V in map reading and analysis, which has a certain degree of innovation. Although the results of the study are relatively preliminary, they are enough to show the great potential of GPT-4V in intelligent map reading. It is conceivable that in the future a new map (or several) will be thrown to GPT-4V, which can give us a very complete and correct enough interpretation. The paper still has the following problems that need to be improved.

1. There are too few maps used in each case (only one map), can we increase the number of maps? and then see how many GPT-4V interpretations are correct and how many are wrong?

2. The source of the map is also a problem, if they all come from the maps in the existing materials, is it a bit unrepresentative? Is it possible to make a series of maps (different types, different countries, different themes, etc.) by yourself to see if GPT-4V can interpret it correctly?

3. Keywords can be optimized, for example, Map Reading can be added

4. There is a question worth considering: if a similar large model is intelligent enough, and we have enough data, do we still need to make a detour to generate a map? After all, the map is for people to see, and if we have the data behind it, we may be able to directly deduce many conclusions (no need to generate a map). But when data is not available, it is valuable to be able to draw some accurate qualitative conclusions from these maps in image format alone.

5. The advantages and disadvantages summarized in Table 1 seem to lack systematization and completeness, and only some contents are listed at random. Could the author please consider how to continue to improve?

6. Generally speaking, the topic of this paper is a bit big, and we should focus on evaluating a specific map reading function of GPT-4V. For example, how often is it correct to answer a specific type of question (e.g., to find an area where a phenomenon is clustered)?

At the same time, there are a few minor errors:

1. In line 47, the AI should write the full name.

2. Line 192, it should be Prompt 2.1, right?

3. What do lines 271-273 mean?

Reviewer 3 Report

Comments and Suggestions for Authors

Very interesting paper specially if it can be published soon. It is a topic of great discussions and this experiment brings some light to some of the issues concerning the use of GPT in GIS.

I liked the paper. Just a few comments

- define in the beginning the nature of your data: what type of images, vector, etc

- In the discussion section you describe advantages and disadvantages. However, all the results you show in the results section are positive. You need also to provide evidence of the disadvantages such as those mentioned in your discussion: precisions, prompt engineering, validation,, etc

Round 2

Reviewer 2 Report

Comments and Suggestions for Authors

The author has revised the manuscript according to the expert's suggestions. It is recommended to accept the manuscript after reviewing the whole text.